# A Deep Long-Term Joint Temporal–Spectral Network for Spectrum Prediction

**DOI:** 10.3390/s24051498

**Published:** 2024-02-26

**Authors:** Lei Wang, Jun Hu, Rundong Jiang, Zengping Chen

**Affiliations:** School of Electronic and Communication Engineering, Sun Yat-sen University, Shenzhen 518107, China; wanglei63@mail2.sysu.edu.cn (L.W.); jiangrd3@mail2.sysu.edu.cn (R.J.); chenzengp@mail.sysu.edu.cn (Z.C.)

**Keywords:** spectrum prediction, long-term joint temporal–spectral network, Bi-ConvLSTM, deep learning

## Abstract

Spectrum prediction is a promising technique to release spectrum resources and plays an essential role in cognitive radio networks and spectrum situation generating. Traditional algorithms normally focus on one-dimensional or predict spectrum values in a slot-by-slot manner and thus cannot fully perceive the spectrum states in complex environments and lack timeliness. In this paper, a deep learning-based prediction method with a simple structure is developed for temporal–spectral and multi-slot spectrum prediction simultaneously. Specifically, we first analyze and construct spectrum data suitable for the model to simultaneously achieve long-term and multi-dimensional spectrum prediction. Then, a hierarchical spectrum prediction system is developed that takes advantage of the advanced Bi-ConvLSTM and the seq2seq framework. The Bi-ConvLSTM captures time–frequency characteristics of spectrum data, and the seq2seq framework is used for long-term spectrum prediction. Furthermore, the attention mechanism is used to address the limitations of the seq2seq framework that compresses all inputs into fixed-length vectors, resulting in information loss. Finally, the experimental results have shown that the proposed model has a significant advantage over the benchmark schemes. Especially, the proposed spectrum prediction model achieves 6.15%, 0.7749, 1.0978, and 0.9628 in MAPE, MAE, RMSE, and $R^{2}$, respectively, which is better than all the baseline deep learning models.

## 1. Introduction

The electromagnetic spectrum, as a critical national strategic resource all over the world, has drawn more and more attention [1,2]. With the development of information technology, the spectrum demand also explodes, but spectrum utilization is low at the same time. As pointed out by the Federal Communications Commission (FCC), numerous allocated spectrum resources are idle to a large extent in time and space [3,4], and the average utilization rate does not exceed 5% at any time and place [5]. In addition, as the spectrum sensing capability of monitoring devices is limited, spectrum monitoring data are very sparse or even scarce in multiple dimensions, such as time, space, and the frequency domain. So, it is difficult to form a detailed and comprehensive spectrum situation, which results in great challenges regarding the full use of the spectrum. Therefore, how to fully grasp the spectrum situation and further improve spectrum utilization are essential to solve the shortage of spectrum resources in the complex electromagnetic environment.

Cognitive radio networks (CRNs) are believed to be an important and promising method to relieve the shortage, whose key technologies include spectrum sensing, spectrum decision, spectrum sharing, spectrum prediction, and spectrum shifting. Figure 1 shows the working process of CRNs; firstly, spectrum sensing, as a key technology of CRNs, can obtain the spectrum state effectively [6,7] to improve spectrum utilization through a large number of continuous detections.

Secondly, the spectrum decision selects the best hole for access and realizes the spectrum shifting according to hole characteristics by spectrum sensing. Then, the spectrum decision provides channel capacity information for spectrum sharing. However, as users increase, the time and energy consumed by spectrum detection increase sharply, which could cause spectrum holes. To solve the above problem, spectrum prediction, as another key technology of CRNs and spectrum situation generation [8], has abstracted increasing attention.

Spectrum prediction utilizes historical spectrum sensing information to mine the occupancy pattern of each channel in the time domain and the potential correlation between channel states and then makes a prediction of channel occupancy in the future time slots. It can quickly find the spectrum holes, guide the spectrum sensing in the next moment and the subsequent spectrum decisionmaking, reduce the amount of spectrum switching, and improve spectrum utilization.

The research on spectrum prediction has made great progress during the past few years. Several traditional spectrum prediction techniques [9], such as regression model [10], Markov model [11], Bayesian inference [12], support vector machine [13], artificial neural network [14], and matrix/tensor completion [15,16], have been proposed for spectrum prediction. Although the traditional methods can improve spectrum utilization to some extent, they are unable to meet the actual needs.

Recently, deep learning has shown its promising capability in spectrum prediction, with high precision, strong robustness, and adaptability. Recurrent neural network (RNN)-based methods (Long Short-Term Memory (LSTM) and gated recurrent unit network (GRU)) [17] are capable of mining underlying temporal correlations among spectrum data. In [18], LSTM was employed to simultaneously predict the Radio Spectrum State (RSS) for two time slots, which requested a large amount of computing resources and suffers from very long training time. To solve the above problems, Ling Yu et al. [19] introduced the Taguchi method and LSTM for time domain spectrum prediction, effectively reducing the time and computational power requirements. Xue Wang et al. [20] used the Back Propagation-LSTM (BP-LSTM) network model for spectrum prediction, which has better prediction performance than BP, LSTM, and GRU. Nevertheless, the spectrum states are interrelated in the time and frequency domains. MA Aygül et al. [21,22] exploited correlation over time and frequency of spectrum data through a 2D-LSTM with better performance than 1D-LSTM, but this method had a weak ability to extract frequency relationships.

To address the aforementioned issues, hybrid networks have good performance in extracting temporal–spectral features of spectrum data by taking advantage of different networks. Lixing Yu et al. [23] combined convolutional neural network (CNN) with the GRU for temporal–spectral spectrum prediction with high prediction accuracy. In [24], a scheme was formulated with CNN-LSTM for multi-dimensional spectrum prediction. Xiaojin Ding et al. [25] combined CNN and bidirectional LSTM (BiLSTM) for predicting the state of multi-channels with higher accuracy than LSTM, BiLSTM, and CNN-LSTM. Graph convolution network (GCN) has a stronger ability to extract spatial features than CNN. Han Zhang et al. [26] designed a graph network model combining GCN and LSTM for multi-channel spectrum prediction that had better predictive performance compared with other methods. To improve prediction accuracy, Xile Zhang et al. [27] proposed a multi-band spectrum prediction method based on attention graph convolutional recurrent neural networks (A-GCRNN), which applied temporal correlation and frequency band correlation to spectrum prediction tasks. These methods have good performance for multi-dimensional spectrum prediction but predict spectrum values in a slot-by-slot manner, thus lacking timeliness.

Overall, simultaneously achieving multi-dimensional and long-term spectrum prediction is a great challenge. Multi-dimensional and long-term studies on effective prediction algorithms are relatively few. Ling Yu et al. [28] constructed a temporal–spectral residual network for multi-slot high-frequency (HF) band prediction from an image inference, but the structure of this method is complex, and selecting data with different time trends is a major challenge. In this study, to solve the above challenging problems, we propose a model with a simple structure for long-term and temporal–spectral spectrum prediction simultaneously. Firstly, we propose a spectrum matrices construction method that contains values of multiple channels at different time slots. Secondly, the seq2seq model is used for long-term spectrum prediction. It performs better in multi-step prediction by considering sequence dependencies between output labels. In addition, it also has a flexible framework. Thirdly, we apply a Bi-ConvLSTM as an encoder to mine the correlations of spectrum values across different channels and extract temporal features within a certain time window. Bi-ConvLSTM is an improved network based on LSTM, which not only enables temporal modeling but also characterizes spectral characteristics [29]. Although the decoder in seq2seq is greatly affected by the length of the intermediate vector, it can be relieved by introducing the attention mechanism. The attention mechanism can solve the limitation of the seq2seq structure, which encodes the constructed spectrum data into a sequence of vectors. In addition, we notice that the correlation between the spectrum states is not equal. Thus, the attention mechanism is also used to assign weights for different spectrum points, which improves the model prediction accuracy. Finally, we utilize Bi-ConvLSTM and dense networks as a decoder, which can adaptively focus on certain parts of spectrum states.

Our main contributions in this paper are the following:We propose a spectrum data construction method with a simple structure to make multi-dimensional and long-term spectrum predictions simultaneously. Different from the existing spectrum prediction in a slot-by-slot manner, the proposed approach is more efficient and can predict multi-slot spectrum states ahead of multiple spectrum points.We combine Bi-ConvLSTM and seq2seq to construct the proposed networks that can achieve both temporal–spectral and long-term spectrum prediction.Validated on real-world datasets, the experimental results show that our proposed spectrum prediction model achieves 6.15%, 0.7749, 1.0978, and 0.9628 in mean absolute percentage error (MAPE), mean absolute error (MAE), root mean square error (RMSE), and R-squared ($R^{2}$), respectively, which is better than all the baseline deep learning models. Furthermore, the designed model is robust against missing spectrum data.

The remainder of this paper is organized as follows. Section 2 compares the differences between the proposed prediction model and traditional prediction models. Section 3 presents the structure of the long-term joint temporal–spectral network. Section 4 shows the experimental results of prediction on real-world spectrum datasets. The Section 5 draws some conclusions.

## 2. Problem Formulation

As shown in Figure 2, most traditional prediction models learn the inherent relationship (T−1) consecutive column vectors {$\chi_{1}$, $\chi_{2}$, …, $\chi_{T-1}$} to predict the spectrum values $\chi_{T}$ at time *T*, and each column vector represents spectrum states of *F* points. Window with fixed length moves forward slot by slot over time and spectrum values. Subsequently, states and qualities of spectrum points in different time slots can be predicted. Considering the actual demand and forecasting timeliness, the prediction model needs to simultaneously predict the values with acceptable error within a relatively long period. Figure 3 shows the principle of the proposed spectrum prediction model; the input and output of the prediction model are {$\chi_{t-n}$, $\chi_{t-n+1}$, …, $\chi_{t}$}, {${\hat{\chi }}_{t-n+1}$, ${\hat{\chi }}_{t-n+2}$,…, ${\hat{\chi }}_{t+1}$}, respectively.

Spectrum data do not exist independently and have closely intrinsic relationships in various dimensions. Therefore, our study utilizes the correlation of spectrum data in the time–frequency domain to research spectrum prediction. The F−T grid diagram is constructed with a spectrum value of *m* spectrum points to every time slot for the proposed spectrum prediction model, as shown in Figure 4. The rows and columns of the grid diagram represent spectrum point *m* and time slot *t*, respectively. There is a correlation between rows and columns in the graph in terms of frequency and time, respectively, and adjacent time slots and channels have a stronger correlation. The spectrum state of *m* spectrum point in each *T* time slot is χ. The process of long-term joint temporal–spectral spectrum prediction can be represented as follows:(1)$$\langle \chi_{t-n},\chi_{t-n+1},\ldots ,\chi_{t-2},\chi_{t-1},\chi_{t}\rangle \to \langle \chi_{t+1}\rangle$$

## 3. Deep Long-Term Joint Temporal–Spectral Network

In this section, we first analyze the proposed model. To achieve time–frequency joint prediction, the proposed model adopts Bi-ConvLSTM to bidirectionally extract temporal and spectral features of historical spectrum data, which can improve accuracy. In addition, the proposed model also adopts seq2seq framework so that the inputs and outputs are matrices to achieve multi-step prediction. The attention mechanism in proposed model assigns weights based on the correlations between different frequency points. Therefore, the proposed model can achieve long-term and temporal–spectral spectrum prediction simultaneously and improve the spectrum prediction performance. We conduct spectrum prediction by developing a hierarchical deep learning framework that consists of an input layer, an encoder layer, a decoder layer, and an output layer, as shown in Figure 5. The specific processing and forecasting process is as follows.

(1) Input layer: Spectrum prediction models can be roughly divided into binary prediction and power level prediction [30]. Different spectrum prediction types have different meanings in specific scenarios. In this study, power spectral density (PSD) is chosen to predict. First, we construct the spectrum matrix for the proposed model. Specifically, denote by $\chi_{t}$ all spectrum points’ values of time length *T*. The current and previous spectrum values are represented as {$\chi_{t-n}$, $\chi_{t-n+1}$, …, $\chi_{t}$}.

(2) Encoder layer: Second, the constructed spectrum data of the input layer {$\chi_{t-n}$, $\chi_{t-n+1}$, …, $\chi_{t}$} are sent to the encoder layer. The encoder layer consists of a Bi-ConvLSTM network and an attention mechanism, which encodes the input data into a vector sequence. The Bi-ConvLSTM contains two convolutional layers and two max-pooling layers in our proposed model. Different from other methods, bidirectional extracts regional temporal and spectral correlation features simultaneously and improves the proposed model’s performance. The structure of the Bi-ConvLSTM and its internal structure are shown in Figure 6 and Figure 7, respectively.

Bi-ConvLSTM: Bi-ConvLSTM achieves forward hidden layer state $\vec{h_{t}}$ and backward hidden layer state $\overset{\leftarrow }{h_{t}}$ that are opposite along the time axis by forward convLSTM and backward convLSTM. The forward convLSTM and backward convLSTM obtain information about the spectrum data along the time axis in the past and future, respectively. The key equations that define the Bi-ConvLSTM network for a given input $\chi_{t}$ are provided as follows:(2)$$\vec{h_{t}}=\vec{ConvLSTM}\left(h_{t-1},\chi_{t},c_{t-1}\right)$$
(3)$$\overset{\leftarrow }{h_{t}}=\overset{\leftarrow }{ConvLSTM}\left(h_{t+1},\chi_{t},c_{t+1}\right)$$
(4)$$h_{t}=\left[\vec{h_{t}},\overset{\leftarrow }{h_{t}}\right]$$
(5)$$\begin{array}{cc} \; & f_{t}=\sigma \left(W_{\chi }^{f}*\chi_{t}+W_{h}^{f}*h_{t-1}+W_{c}^{f}⊙c_{t-1}+B_{f}\right)\\ \; & i_{t}=\sigma \left(W_{\chi }^{i}*\chi_{t}+W_{h}^{i}*h_{t-1}+W_{c}^{i}⊙c_{t-1}+B_{i}\right)\\ \; & {\tilde{c}}_{t}=h\left(W_{\chi }^{c}*\chi_{t}+W_{h}^{c}*h_{t-1}+B_{c}\right)\\ \; & c_{t}=i_{t}⊙{\tilde{c}}_{t}+f_{t}⊙c_{t-1}\\ \; & o_{t}=\sigma \left(W_{\chi }^{o}*\chi_{t}+W_{h}^{o}*h_{t-1}+W_{c}^{o}⊙c_{t}+B_{o}\right)\\ \; & h_{t}=o_{t}⊙\hbar \left(c_{t}\right) \end{array}$$
where $\chi_{t}$ represents input spectrum data. $W_{\chi }^{f}$, $W_{\chi }^{i}$, and $W_{\chi }^{o}$ are weight matrices of input, hidden state cell, and memory cell in forgotten gate, respectively. $W_{h}^{f}$, $W_{h}^{i}$, and $W_{h}^{f}$ are weight matrices of input, hidden state cell, and memory cell in input gate, respectively. $W_{c}^{f}$, $W_{c}^{i}$, and $W_{c}^{o}$ are the weight matrices of input, hidden state cell, and a memory cell in the output gate, respectively. $B_{f}$, $B_{i}$, $B_{c}$, and $B_{0}$ are bias of forgotten gate, input gate, current state, and output gate, respectively. $c_{t-1}$ and ${\tilde{c}}_{t}$ are long-term memory and current memory. $h_{t}$ is hidden state, which contains $\vec{h_{t}}$ and $\overset{\leftarrow }{h_{t}}$. ⊙ and * are Hadamard products and convolutional operators, respectively. σ, *h* are sigmoid and tanh activation functions, respectively.

Attention Mechanism: The relationships of each spectrum point are set equally in the traditional seq2seq model. However, the degree of correlation between channels is different. So, it is not appropriate to set an equal weight value for each set of input spectrum data. Therefore, the attention mechanism is employed to solve the above drawbacks and reserves more information of input spectrum data. The attention layer in this proposed structure consists of two parts for each input, the outputs of all Bi-ConvLSTM units in the encoder and the input state of a corresponding Bi-ConvLSTM in the decoder. In particular, the internal structure of the attention layer is shown in Figure 8. The alignment model $e_{ij}$ is calculated by Dot. $s_{i}$ is the hidden state of the neural network in the decoder at time *i*. $h_{j}$ is the *j*-th corresponding hidden vector of the input sequence in the encoder.

(3) Decoder layer: Then, the vectors generated by the encoder layer are sent to the decoder layer. The decoder layer consists of a Bi-ConvLSTM network and a dense network. It adaptively selects a subset of these vectors as the decoding result based on the intermediate vectors obtained by the encoder and the historical parameters in the network. Similarly, the Bi-ConvLSTM of the decoder layer contains two convolutional layers and two max-pooling layers.

A Dense Network: The dense network is employed to convert the vector dimension of output from the Bi-ConvLSTM of the decoder layer to the dimension of the final predicted value, which has several layers of fully connected neural nodes. Specifically, the dense network consists of three full layers of fully connected neural nodes in our proposed model.

(4) Output layer: Finally, the output of the dense layer is sent to an activation function to obtain a prediction result. The fina output is $\chi_{t+1}$, which donates future values of all spectrum points at multiple time slots.

Activation function and loss function: In particular, the prediction result at a time slot *T* and the real spectrum states at its next slot are sent to a loss function to predict spectrum values at the next time slot by updating the gradients calculated by the loss function. The gradients are sent to the Bi-ConvLSTM network and dense network for updating their weight matrices to decrease the distance between the predicted result and the ground truth. To find a final prediction result, we choose a tanh function as the activation function that takes the output of the dense network as the input that maps the prediction result into a vector of elements between −1 and 1. In addition, we choose the mean squared error (MSE) as the loss function of the proposed model, which is widely applied in deep learning applications.

Training algorithm: The proposed spectrum prediction model is trained by minimizing MSE between the true value and the predicted value as
(6)$$L\left(w,b\right)={∥\chi_{t+1}-{\hat{\chi }}_{t+1}∥}^{2}$$
where w, b are all learn-able parameters in the proposed spectrum prediction model.

Algorithm 1 outlines the proposed model training process. Firstly, training sequences are constructed from the original spectrum data (lines 1–3). Then, forward propagation and back-propagation are repeatedly applied to train the model (lines 4–14).
**Algorithm 1** The proposed model training algorithm.**Input:** Historical spectrum data sequence**Output:** Trained long-term joint temporal–spectral network//Construct the dataset1: construct spectrum matrix $\chi_{t-n},\chi_{t-n+1},\ldots ,\chi_{t-2},\chi_{t-1},\chi_{t}$ from historical spectrum data2: *D*←ϕ3: divide set *D* into train set $D_{train}$ and test set $D_{test}$4: //Train the proposed prediction model5: initialize all learn-able parameters in long-term joint temporal–spectral network6: **Repeat**7:      randomly select a batch of instances from $D_{train}$8:      find w, b by minimizing the objective (6)9: **Until** the training epochs are met10: //Test the proposed prediction model11: **for** each sample in $D_{test}$ **do**12:      fed into the trained proposed model13:      output the prediction results of that sample14: **end for**

## 4. Experiments

### 4.1. Settings

Dataset Description: In this section, we design some comparative experiments to evaluate the performance of the proposed spectrum prediction method. The datasets used in our spectrum prediction experiments are gathered from RWTH Aachen University in Germany [31]. These datasets are available at https://github.com/chengrunmeng/Aachen-spectrum-data-part accessed on 2 July 2023. Specifically, the data are about spectrum states measured at the residential area in Maastricht. The datasets consist of two sub-bands with 770 MHz and 3750 MHz central frequencies and are named dataset 1 and dataset 2, respectively. Each sub-band has a bandwidth of 1500 MHz and a frequency resolution of 200 kHz.

Hyper-parameters: The settings of the proposed model are as follows. ConvLSTM2D **1** is composed of 5 identical convolution kernels with the size 5 × 5, while ConvLSTM2D **2** consists of 10 kernels with the size 3 × 3. Particularly, the convolution kernels slide over full rows of the input data matrix to extract spectral correlation features at a multi-time slot. The number of hidden units is 128. Adam is used as the optimizer. The number of epochs and initial learning rate are set as 200 and 0.001, respectively. The batch size is 32. The length of time is 30, which is the most recent data that we consider as input to predict the values of all the spectrum points at the 30 time slots. We conduct extensive experiments to test the proposed model by varying the time length ranging from 10 to 50 and find that setting 30 can achieve the best performance. MSE is used as a loss function for proposed model training and testing. The prediction time slots are 30. In addition, 90% of the spectrum data are used to train the model and treat the remaining 10 % as the test set.

Baselines and Evaluation Metrics: For comparison, 10 prediction models based on deep learning are considered as baselines, which are LSTM [18], GRU [32], BiLSTM [33], CNN [34], CNN-LSTM [35], CNN-BiLSTM [25], CNN-BiLSTM-attention [36], seq2seq-LSTM [37], seq2seq-LSTM-attention [38], and seq2seq-Bi-ConvLSTM [39]. In order to fairly compare the performance of models, the input information and output information of all prediction models are all the same and the baseline models use the multi-step principle to output results. Moreover, to evaluate the performance of the designed model, MAPE, MAE, RMSE, and $R^{2}$ are used to evaluate the performance. Especially, the calculation process is as follows:(7)$$MAPE=\frac{100\%}{N}\sum_{k=1}^{N}\left|\frac{{\hat{y}}_{k}-y_{k}}{y_{k}}\right|$$
(8)$$MAE=\frac{1}{N}\sum_{k=1}^{N}\left|\;y_{k}-{\hat{y}}_{k}\right|$$
(9)$$RMSE=\sqrt{\frac{1}{N}\sum_{k=1}^{N}{\left(y_{k}-{\hat{y}}_{k}\right)}^{2}}$$
(10)$$R^{2}=1-\frac{\sum_{k=1}^{N}{\left(y_{k}-{\hat{y}}_{k}\right)}^{2}}{\sum_{k=1}^{N}{\left(y_{k}-{\bar{y}}_{k}\right)}^{2}}$$
where $y_{k}$ is the ground truth, ${\hat{y}}_{k}$ is the prediction value, and ${\bar{y}}_{k}$ is the average value of the ground truth.

### 4.2. Experimental Results and Discussion

Effect of The Time Length: Choosing the appropriate time length is crucial for improving model performance.

So, we first explore the effect of time length on the performance of the proposed model and select the most appropriate values as parameters. Figure 9 shows the impact of the time length on the performance of the proposed model. In this experiment example, we vary the time length and measure the accuracy achieved by the designed model. The value of the accuracy metric increases with the increase in the time length. When the time length is 30, the accuracy is highest. Then, the value is decreasing when the time length grows. This result shows that the best time length is about 30 for our designed model and the used dataset.

Baseline Model Comparison: Figure 10 shows the performance of the designed model regarding dataset 1 and dataset 2, respectively. From Figure 10, we can see that the designed model can achieve better performance than CNN-BiLSTM-attention and seq2seq-LSTM-attention on two spectrum datasets. Furthermore, from the above result, it can also be seen that the proposed model has good generalization ability. The reason for choosing the above two models for comparison is as follows. According to the results of the literature [25], CNN-BiLSTM-attention achieves better performance for time–frequency spectrum prediction than LSTM, BiLSTM, GRU, and CNN-LSTM. In addition, seq2seq-LSTM-attention has good performance on long-term time series prediction.

Table 1 demonstrates the performance improvement by comparing the designed model with ten baseline models. Firstly, the spectral and temporal correlation features are hidden in the spectrum data. We observe that the learning models with the CNN are better than that only considering the temporal correlations. Secondly, hybrid models (such as CNN-LSTM) perform better than single models (such as LSTM, GRU, BiLSTM, and CNN), which is because hybrid models take advantage of different networks. Then, the models with bidirectional structures (such as BiLSTM and CNN-BiLSTM) perform better than one-directional models (such as LSTM and CNN-LSTM), which is due to the fact that the models with bidirectional structures can extract bidirectional features of spectrum data. While these models perform better, they are more complex in structure and more difficult to train. In addition, the models with an attention mechanism (such as CNN-BiLSTM-attention and seq2seq-LSTM-attention) perform better than the models without an attention mechanism (such as CNN-BiLSTM and seq2seq-LSTM), which is because the attention mechanism focuses on the correlation between spectrum values of different frequency points at each moment in time to assign weights for the vector set. Finally, the proposed model, advanced Bi-ConvLSTM and seq2seq structure, can achieve the best performance regarding MAE, MAPE, RMSE, and $R^{2}$. Take the MAE as an example: LSTM, GRU, BiLSTM, CNN, CNN-LSTM, CNN-BiLSTM, CNN-BiLSTM-attention, seq2seq-LSTM, seq2seq-LSTM-attention, seq2seq-Bi-ConvLSTM, and seq2seq-Bi-ConvLSTM-attention achieve MAE values of 0.8982, 0.8847, 0.8796, 0.8679, 0.8533, 0.8498, 0.8345, 0.8556, 0.8436, 0.8064, and 0.7749, respectively. So, the proposed model can effectively reduce the amount of spectrum switching and improve the spectrum utilization through simultaneous multi-step and joint time–frequency prediction.

The Stability of the Proposed Model: Figure 11 demonstrates the stability of the designed model when using it in datasets with missing data. In particular, we use the spectrum data with different missing rates to evaluate the performance of the designed model.

Firstly, the performance of all the prediction models decreases as the rate of missing data increases, which is because, the greater the rate of missing data, the more information is lost. Secondly, we can find that the designed model can achieve the highest accuracy when the spectrum datasets are incomplete with missing data. The proposed model, CNN-BiLSTM-attention, and seq2seq-LSTM-attention can achieve MAE values of 0.7864, 0.8543, and 0.8876, respectively, when the missing ratio is 5% in dataset 1. Similarly, the proposed model also performs best on dataset 2 with different missing rates. In particular, the decreasing performance slope of the proposed model is the lowest when the missing rate is increasing from 5% to 10% and 15%. This also shows the strong stability of the designed model.

## 5. Conclusions

In this paper, we have investigated the problem of spectrum occupancy prediction and proposed a deep learning prediction model achieving both long-term and time–frequency spectrum prediction. To be more specific, we take advantage of advanced Bi-ConvLSTM to extract temporal–spectral relationships in spectrum data and the seq2seq framework for long-term spectrum prediction. Then, the attention mechanism is used to solve the seq2seq limitation. Subsequently, we evaluate the prediction performance of the conventional LSTM, GRU, BiLSTM, CNN, CNN-LSTM, CNN-BiLSTM, CNN-BiLSTM-attention, seq2seq-LSTM, seq2seq-LSTM-attention, and seq2seq-Bi-ConvLSTM for comparison purposes. Moreover, the experimental results demonstrate that the proposed model performs better and also is robust to missing data in spectrum training data. Finally, the proposed model has been applied to our program and will be adopted in real-world scenarios.

## Figures and Tables

**Figure 1 sensors-24-01498-f001:**
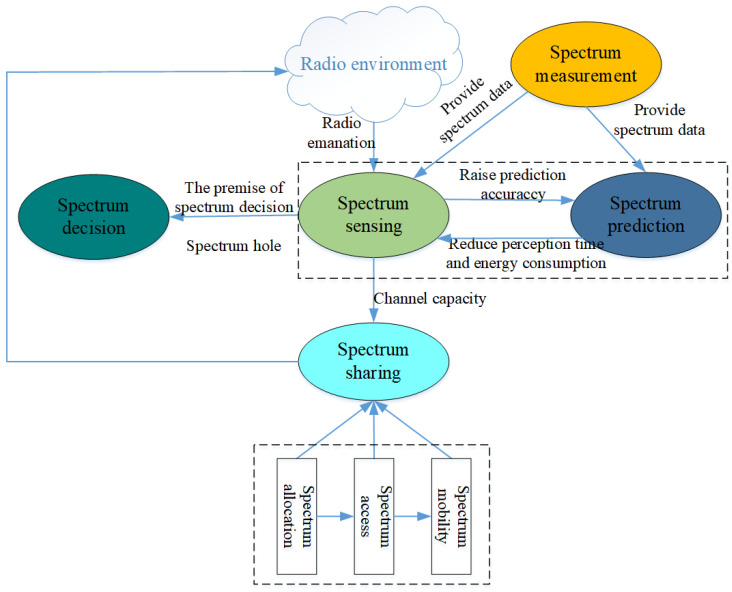
The working process of CRNs.

**Figure 2 sensors-24-01498-f002:**
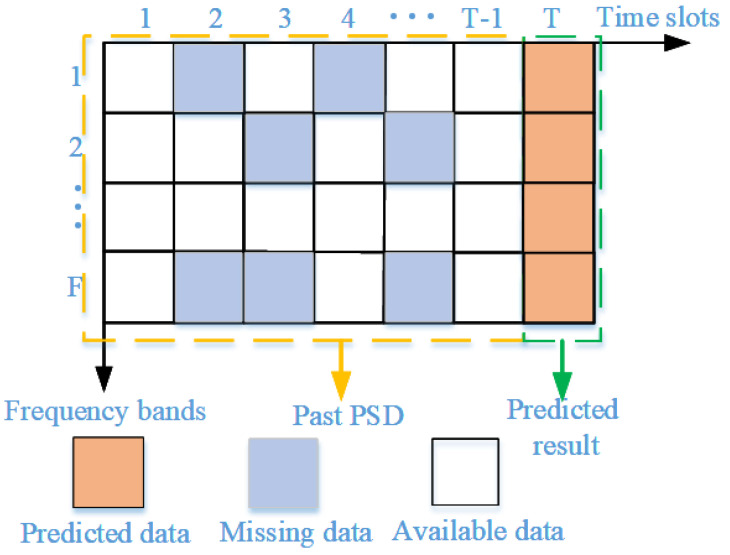
The principle of traditional spectrum prediction models.

**Figure 3 sensors-24-01498-f003:**
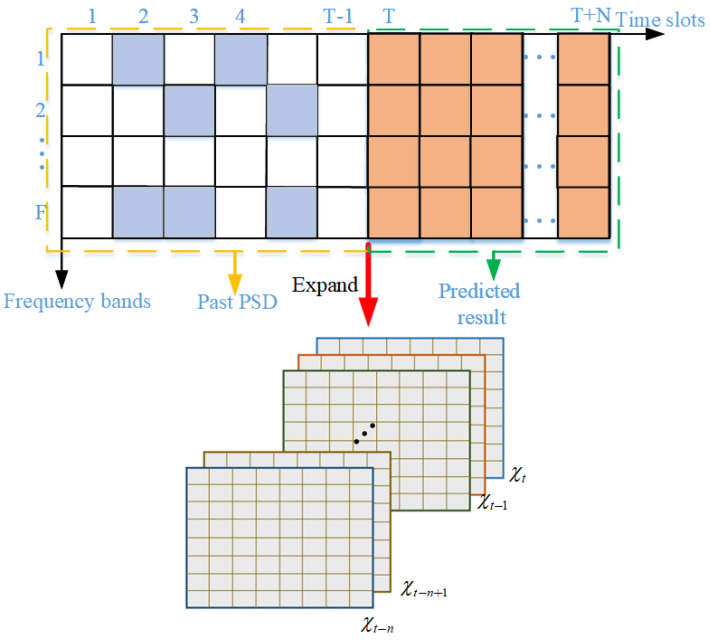
The principle of proposed spectrum prediction model.

**Figure 4 sensors-24-01498-f004:**
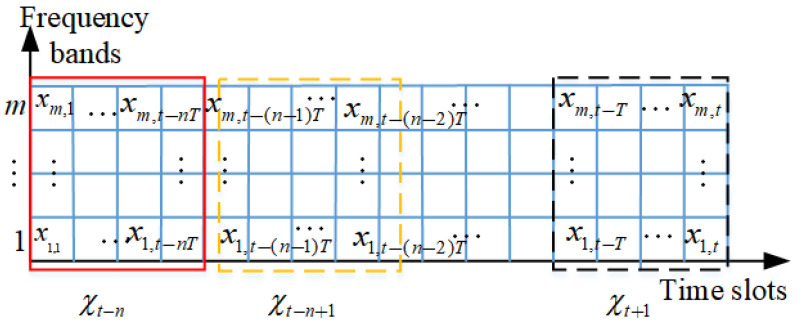
The construction of spectrum data for the proposed prediction model.

**Figure 5 sensors-24-01498-f005:**
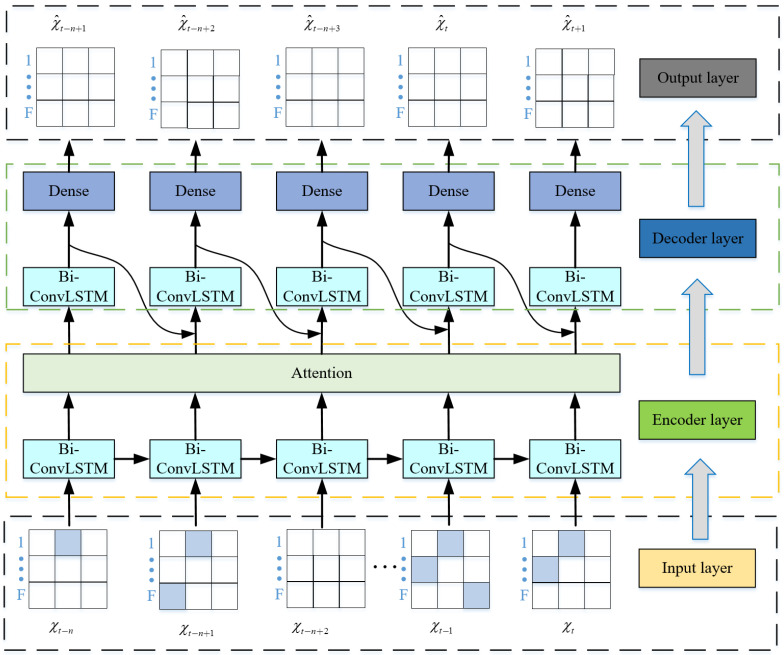
The construction of long-term joint temporal–spectral spectrum prediction models.

**Figure 6 sensors-24-01498-f006:**
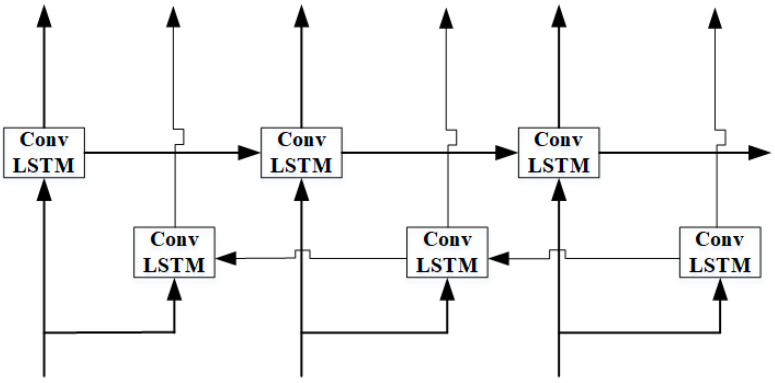
The structure of Bi-ConvLSTM.

**Figure 7 sensors-24-01498-f007:**
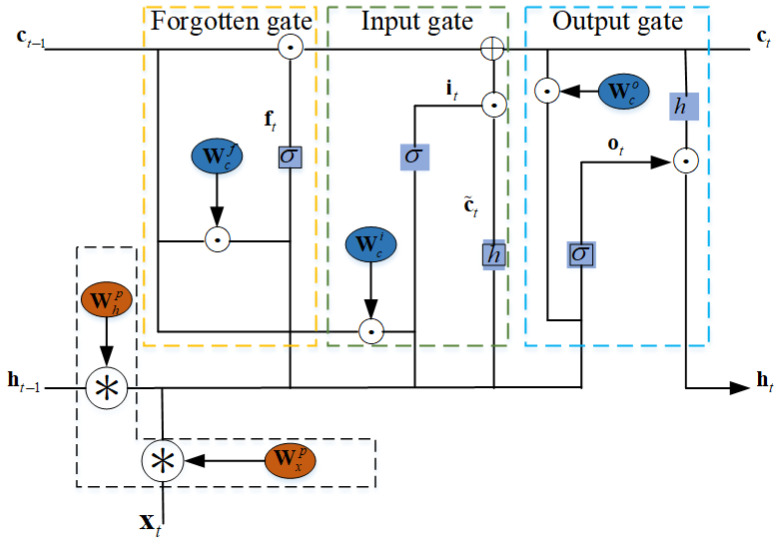
Internal structure of the ConvLSTM.

**Figure 8 sensors-24-01498-f008:**
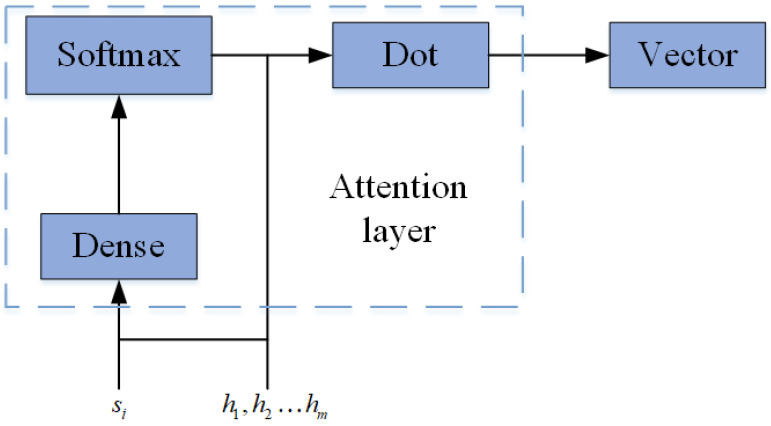
The internal structure of the attention layer.

**Figure 9 sensors-24-01498-f009:**
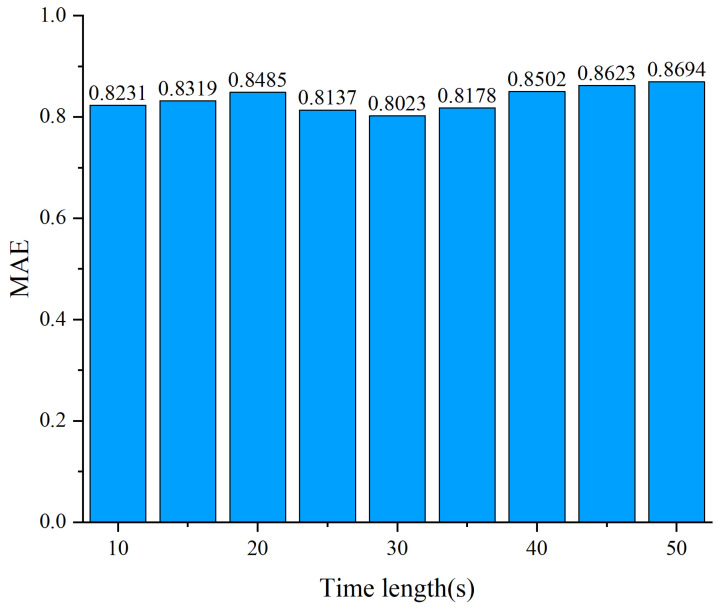
The effect of the time length of the designed model.

**Figure 10 sensors-24-01498-f010:**
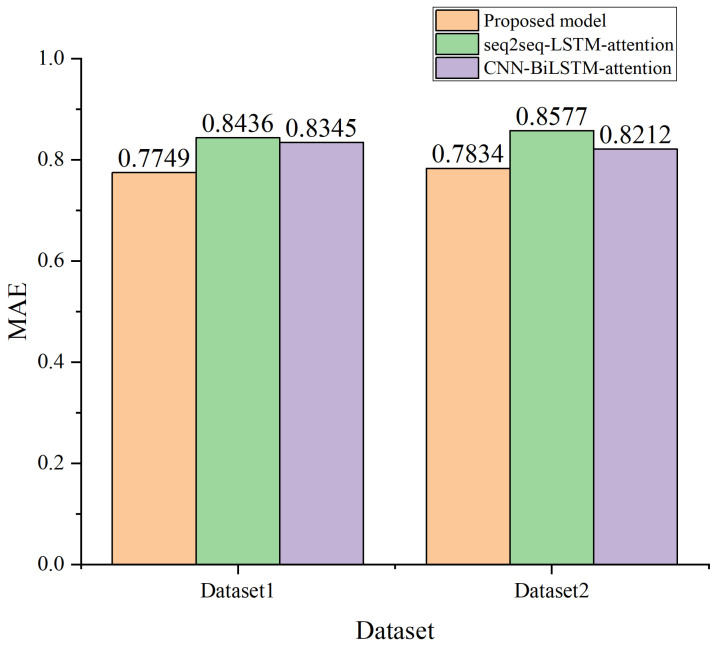
Performance comparison with the existing schemes.

**Figure 11 sensors-24-01498-f011:**
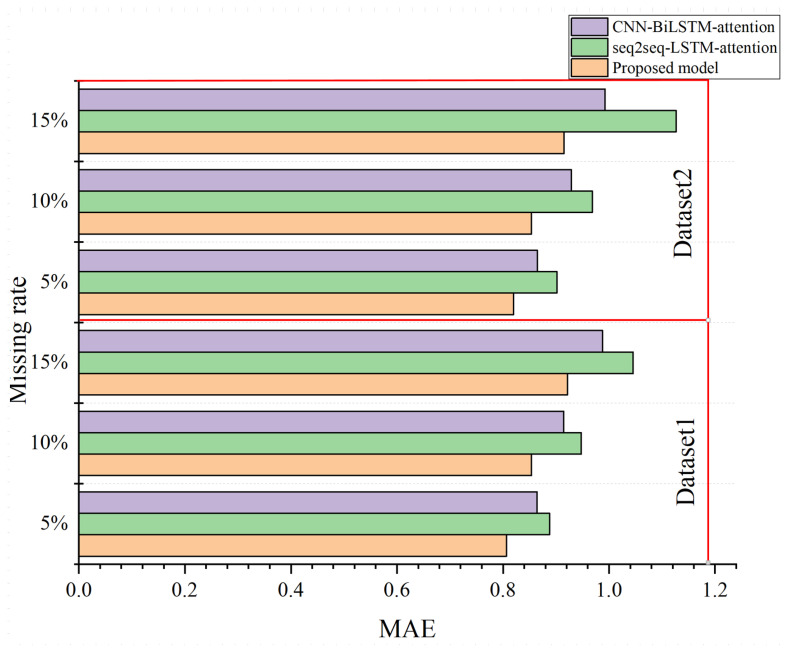
The stability against missing data for spectrum prediction.

**Table 1 sensors-24-01498-t001:** The comparison of performance on test set.

Methods	MAE	MAPE	RMSE	$R^{2}$
LSTM	0.8982	8.23%	1.6512	0.8223
BiLSTM	0.8796	8.07%	1.5728	0.8473
GRU	0.8847	8.17%	1.5851	0.8398
CNN	0.8679	7.93%	1.5695	0.8491
CNN-LSTM	0.8533	7.67%	1.5588	0.8587
CNN-BiLSTM	0.8498	7.45%	1.5537	0.8619
CNN-BiLSTM-attention	0.8345	7.25%	1.4218	0.9065
Seq2seq-LSTM	0.8556	7.47%	1.5618	0.8578
Seq2seq-LSTM-attention	0.8436	7.36%	1.5465	0.8783
Seq2seq-Bi-ConvLSTM	0.8064	6.64%	1.2355	0.9362
Seq2seq-Bi-ConvLSTM-attention	0.7749	6.15%	1.0978	0.9628

## Data Availability

The data that support the findings of this study are openly available at https://github.com/chengrunmeng/Aachen-spectrum-data-part.

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
