# Peer review of "A Deep Long-Term Joint Temporal–Spectral Network for Spectrum Prediction"

_sensors, 2024, doi:10.3390/s24051498_

Round 1

Reviewer 1 Report

Comments and Suggestions for Authors

In this manuscript, the authors propose a time-frequency and multi-slot spectrum prediction method based on deep learning. In specific, the authors formulate the problem similar to time-series prediction problems (e.g., using RNNs), using PSD values of previous time instances in all the frequency bands to estimate the PSD per frequency band at the next time instance. They describe the data pre-processing and the input/output values, as well as the prediction system involving the Bi-ConvLSTM and the seq2seq frameworks. Furthermore, the authors demonstrate the effectiveness of their method via experimental results, quantifying its performance with 4 metrics and comparing the results to baseline deep learning models. The paper is in general well-written, interesting and relevant to the topics of Sensors. Some minor comments:

You should conduct a brief complexity analysis of your model.

What is the time required for training (e.g., per epoch)?

You should mention the inference latency of your model compared to the baselines.

In my experience, apart from the “time-window” parameter n that you fine-tuned, there is also the future time instance that you aim to predict the PSD. I see that you aim to predict the PSD at t+1. It would be interesting to mention if it is possible to train the model for t+2 or in general t+m prediction. Relevant references are:

1.       "Proactive autoscaling for edge computing systems with kubernetes." Proceedings of the 14th IEEE/ACM International Conference on Utility and Cloud Computing Companion. 2021.

2.       "A machine learning-driven approach for proactive decision making in adaptive architectures." 2019 IEEE international conference on software architecture companion (ICSA-C). IEEE, 2019.

3.       "Intelligent Mission Critical Services over Beyond 5G Networks: Control Loop and Proactive Overload Detection." 2023 International Conference on Smart Applications, Communications and Networking (SmartNets). IEEE, 2023.

4.       "Achieving predictive and proactive maintenance for high-speed railway power equipment with LSTM-RNN." IEEE Transactions on Industrial Informatics 16.10 (2020): 6509-6517.

Reviewer 2 Report

Comments and Suggestions for Authors

A very good and interesting study done on a correct scientific basis.

Line 151: methods, It bidirectional

it should be lowercase.

Line 210: Data Prepossessing:

should be preprocessing?

Figure 9: Time Length in seconds [s]? put it on Fig. 9?

Line 308: that it performs better

Please define it, which model in particular?

It will be useful that the authors explain the difference between Bi-LSTM and seq2seq since both are RNN.

Reviewer 3 Report

Comments and Suggestions for Authors

This paper presents a deep long-term joint temporal-spectral network for spectrum prediction.

The topic is important and interesting. 

However, this paper contains a number of issues to be solved.

1) Despite the performance advantages of the proposed method, the authors should further elaborate on the rationale of the proposed method. For example, why can the proposed method improve the spectrum sensing prediction? 

2) A number of writing issues can be found in the paper.

   2.1) the full terms of acronyms, such as MAPE, MAE, and RMSE should be given when they first appear in the text (not in the experimental section).

   2.2) Figure. xx should be Figure xx, where there should no ".".

   2.3) There should be a white space between citation and the text.

   2.4) "The structure of the Bi-ConvLSTM and its’ internal structure are shown in Figure.6 and Figure.7" -> "The structure of the Bi-ConvLSTM and its’ internal structure are shown in Figure 6 and Figure 7, respectively".

   2.5) The caption of Figure 1 is confusing. What is the meaning of "key link"?

   2.6) You should place figures along with the corresponding texts. 

3) Some technical details should be given in Section 4. For example, in which machine, do you train your models?

The dataset URL should be given (instead of only listing reference [27]).

4) How do you compare your proposed model with baselines in a fair way?

5) Most of the references are quite outdated (before or around 2020). The authors should conduct a more comprehensive literature survey and include some of the latest articles. 

6) The authors should further discuss how the proposed model to be adopted in real-world scenarios. For example, you may consider deploying your model in an edge computing environment. For your reference, you may refer to "Edge-Based Communication Optimization for Distributed Federated Learning, IEEE Transactions on Network Science and Engineering".

Comments on the Quality of English Language

This paper presents a deep long-term joint temporal-spectral network for spectrum prediction.

The topic is important and interesting. 

However, this paper contains a number of issues to be solved.

1) Despite the performance advantages of the proposed method, the authors should further elaborate on the rationale of the proposed method. For example, why can the proposed method improve the spectrum sensing prediction? 

2) A number of writing issues can be found in the paper.

   2.1) the full terms of acronyms, such as MAPE, MAE, and RMSE should be given when they first appear in the text (not in the experimental section).

   2.2) Figure. xx should be Figure xx, where there should no ".".

   2.3) There should be a white space between citation and the text.

   2.4) "The structure of the Bi-ConvLSTM and its’ internal structure are shown in Figure.6 and Figure.7" -> "The structure of the Bi-ConvLSTM and its’ internal structure are shown in Figure 6 and Figure 7, respectively".

   2.5) The caption of Figure 1 is confusing. What is the meaning of "key link"?

   2.6) You should place figures along with the corresponding texts. 

3) Some technical details should be given in Section 4. For example, in which machine, do you train your models?

The dataset URL should be given (instead of only listing reference [27]).

4) How do you compare your proposed model with baselines in a fair way?

5) Most of the references are quite outdated (before or around 2020). The authors should conduct a more comprehensive literature survey and include some of the latest articles. 

6) The authors should further discuss how the proposed model to be adopted in real-world scenarios. For example, you may consider deploying your model in an edge computing environment. For your reference, you may refer to "Edge-Based Communication Optimization for Distributed Federated Learning, IEEE Transactions on Network Science and Engineering".
